# Cold and Exercise: Therapeutic Tools to Activate Brown Adipose Tissue and Combat Obesity

**DOI:** 10.3390/biology8010009

**Published:** 2019-02-12

**Authors:** Carmem Peres Valgas da Silva, Diego Hernández-Saavedra, Joseph D. White, Kristin I. Stanford

**Affiliations:** 1Dorothy M. Davis Heart and Lung Research Institute, The Ohio State University Wexner Medical Center, Columbus, OH 43210, USA; Carmem.PeresValgasDaSilva@osumc.edu (C.P.V.d.S.); Diego.Hernandez-Saavedra@osumc.edu (D.H.-S.); joeywhite2@gmail.com (J.D.W.); 2Department of Physiology and Cell Biology, The Ohio State University Wexner Medical Center, Columbus, OH 43210, USA

**Keywords:** brown adipose tissue, cold, exercise, glucose, lipids, phospholipids, 12,13-diHOME, FGF21, VEGF, obesity

## Abstract

The rise in obesity over the last several decades has reached pandemic proportions. Brown adipose tissue (BAT) is a thermogenic organ that is involved in energy expenditure and represents an attractive target to combat both obesity and type 2 diabetes. Cold exposure and exercise training are two stimuli that have been investigated with respect to BAT activation, metabolism, and the contribution of BAT to metabolic health. These two stimuli are of great interest because they have both disparate and converging effects on BAT activation and metabolism. Cold exposure is an effective mechanism to stimulate BAT activity and increase glucose and lipid uptake through mitochondrial uncoupling, resulting in metabolic benefits including elevated energy expenditure and increased insulin sensitivity. Exercise is a therapeutic tool that has marked benefits on systemic metabolism and affects several tissues, including BAT. Compared to cold exposure, studies focused on BAT metabolism and exercise display conflicting results; the majority of studies in rodents and humans demonstrate a reduction in BAT activity and reduced glucose and lipid uptake and storage. In addition to investigations of energy uptake and utilization, recent studies have focused on the effects of cold exposure and exercise on the structural lipids in BAT and secreted factors released from BAT, termed batokines. Cold exposure and exercise induce opposite responses in terms of structural lipids, but an important overlap exists between the effects of cold and exercise on batokines. In this review, we will discuss the similarities and differences of cold exposure and exercise in relation to their effects on BAT activity and metabolism and its relevance for the prevention of obesity and the development of type 2 diabetes.

## 1. Introduction

The overall prevalence of obesity has increased dramatically over the last several decades [1,2]. The World Health Organization (WHO) has reported that more than 1.9 billion adults around the world are overweight and nearly one-third of the population is obese [3]. Obesity is a consequence of an altered energy balance in which energy intake exceeds energy expenditure [4,5]. This imbalance results in an increased accumulation of adipose tissue and impairments in glucose and lipid metabolism [6,7,8]. Obesity is strongly associated with several comorbidities including type 2 diabetes (T2DM), cardiovascular disease, certain cancers [9,10], and an increased risk of mortality [11], thus, there is a great need for effective therapies to prevent and treat obesity and its associated comorbidities [12]. 

Brown adipose tissue (BAT) is an important target to combat obesity and metabolic disease. BAT is a thermogenic tissue that consumes substantial amounts of glucose and fatty acids as fuel for thermogenesis and energy expenditure [13,14,15]. BAT is innervated by both sympathetic and sensory nerves and is characterized by small, multilocular lipid droplets, a high number of mitochondria, and an abundant expression of the mitochondrial protein, uncoupling protein 1 (UCP1). When activated, UCP1 uncouples mitochondrial respiration from adenosine-5′-triphosphate (ATP) synthesis [16,17], increasing the proton leak across the inner mitochondrial membrane and releasing the proton motive force as heat rather than driving ATP synthase [18]. BAT maintains a high oxidative metabolic capacity, and when it is highly active, it exerts beneficial metabolic effects on obesity, insulin resistance, and atherosclerosis [19,20,21].

There are several established mechanisms that influence BAT activity and metabolism, including cold exposure and exercise [22,23]. These two stimuli have distinct effects on BAT. Cold exposure is the most well-studied means to activate BAT, as the primary role of BAT is to mediate non-shivering thermogenesis in mammals [22]. Cold exposure leads to the activation of the sympathetic nervous system (SNS), causing release of norepinephrine from sympathetic nerves and activation of the β-adrenergic receptor (β-AR). This stimulates cyclic adenosine monophosphate (cAMP)-dependent signaling pathways and results in increased fuel uptake and oxidation for heat generation by UCP1 [24]. Thus, cold-activated BAT increases excess fuel uptake and energy expenditure that likely impacts obesity and insulin resistance.

Exercise enhances insulin sensitivity, improves glucose tolerance, and reduces circulating lipids, all of which improve metabolic health. Exercise causes adaptations to several tissues in the body including skeletal muscle, the cardiovascular system, white adipose tissue (WAT), and BAT [25,26,27,28]. The effect of exercise on BAT, however, has provided some conflicting data with some studies indicating an increase in BAT activity [27,29,30,31], and some demonstrating a decrease in BAT activity [32,33,34]. In contrast to cold exposure, exercise itself is a thermogenic activity, so it is unlikely that exercise would further increase the thermogenic function of BAT [35,36]. The SNS is also stimulated by exercise, so it is possible that exercise-induced β-adrenergic receptor stimulation activates BAT, particularly given that recent studies have demonstrated that exercise is capable of increasing the sympathetic tone and vascularization of BAT [37,38]. Although this is an interesting hypothesis, it has not been experimentally investigated [37]. While the thermogenic activity of BAT is likely not the primary response to exercise, there are other exercise-induced adaptations to BAT that may contribute to the beneficial effects on metabolic health. 

Thus, in this review, we will discuss the current literature examining the effects of cold exposure and exercise on BAT activation and energy expenditure with a focus on how they impact glucose metabolism, lipid metabolism, structural lipids, and BAT-secreted factors (batokines) in terms of their similarities, differences, and potential relevance for the prevention of obesity and T2DM (Figure 1).

## 2. The Role of BAT in Glucose Metabolism in Response to Cold and Exercise

Among the many fuel sources of BAT, glucose is a major contributor and plays an important role in BAT homeostasis [20,22]. Other fuel sources for BAT include dietary and non-esterified fatty acid (FA), as well as glutamate, which can directly or indirectly stimulate thermogenesis [15]. Activated BAT can use glucose and fatty acids as fuel for thermogenesis, enhancing overall energy expenditure in rodents and humans [19,23], and increased BAT activity is associated with lower blood glucose levels in humans [39]. Cold exposure and exercise have distinct effects on glucose uptake and metabolism in BAT; in the section below, we will discuss these effects. 

### 2.1. Cold Exposure Increases Glucose Metabolism in BAT

Cold exposure is a powerful stimulus to increase glucose uptake in BAT. Both acute (4–48 h) and chronic (10 days) cold exposure increased glucose uptake and improved insulin sensitivity by increasing the glucose transporter type 4 (GLUT4) expression or clearing serum triacylglycerol (TAG) in BAT in rodents [19,40,41,42,43]. This increase in glucose uptake is largely due to increased expression of genes and proteins involved in glucose metabolism and insulin signaling. A previous study showed that in a pathophysiological condition, 24-h cold exposure stimulates BAT to clear TAG from circulation, which is associated to an insulin-resistant state [19] (Table 1). Cold exposure also increased glucose uptake in BAT of obese, glucose-intolerant mice (hypertriglyceridemic *Apoa5* deficient mice). The cold-activated increase in BAT glucose uptake was greater than in all other tissues (brain, heart, liver, WAT, and skeletal muscle) combined [19,43]. Together these data indicate that cold exposure is a powerful stimulus to increase glucose uptake and insulin sensitivity in BAT (Figure 1).

In humans, BAT activity is measured by fluorodeoxyglucose (^18^FDG) uptake by positron emission tomography-computed tomography (PET-CT). Studies in humans have shown that cold exposure for 2 h (16 °C–18 °C) [44,45,46], 5–8 h [47], or 10 days [48,49] all increase glucose uptake in BAT (Table 2). In fact, human studies have shown that cold-activated glucose uptake in BAT is greater than insulin-stimulated glucose uptake in BAT [45]; cold exposure increased glucose uptake in BAT up to 12-fold, while insulin stimulation increased glucose uptake only 5-fold in BAT [45]. The increase in cold-activated BAT glucose uptake is correlated with improved whole-body glucose disposal and insulin sensitivity in healthy adults [44], obese adults [49,50], and in patients with T2DM [49]. These data indicate that acute and chronic cold exposure increase glucose uptake and insulin signaling in BAT in both rodents and humans (Figure 1).

### 2.2. Exercise Has Conflicting Effects in Glucose Metabolism in BAT

Exercise is an effective therapy to improve glucose metabolism and insulin sensitivity in skeletal muscle [80,81], heart [82], and WAT [28,83]. Since BAT plays an important role in regulating glucose metabolism, several studies have investigated how exercise can affect glucose uptake in BAT [29,33,72,73,79] (Table 1). 

Some rodent studies have shown that exercise (treadmill training or swimming) [29,72] increases the expression of proteins related to insulin signaling in BAT. This would indicate an increase in glucose uptake into BAT, but that was not directly measured in these studies. In contrast, separate studies have determined that GLUT1 and 4 are not changed in BAT in response to exercise [38,73]. These differences in outcomes for exercise and BAT activity might be due in part to the intrinsic differences in exercise modalities, duration, experimental challenges, rodent model or strain used, and measurement of BAT activity. Additionally, a recent study from our laboratory investigated the effects of 3 weeks of wheel-cage running on glucose metabolism in mice. Exercise increased the expression of genes involved in glucose metabolism in BAT, but brown adipocytes differentiated from the stromal vascular fraction (SVF) isolated from exercise-trained BAT had decreased basal glucose uptake compared to brown adipocytes differentiated from sedentary BAT [73] (Table 1). It is important to note that in vitro conditions seldom include sympathetic innervation and lack neural innervation and vascular supply, the absence of which could impact BAT function. Our results show that mature brown adipocytes from exercise-trained BAT, in the absence of innervation, have reduced glucose uptake.

In humans, recent studies have investigated the effects of exercise on glucose uptake in BAT. One study used ^18^FDG-PET-CT to examine the effects of short-term, high-intensity interval training and moderate-intensity continuous training in sedentary, healthy, male subjects. Interestingly, exercise training decreased insulin-stimulated glucose uptake in BAT [79] (Table 2). Other studies examined the effects of endurance exercise on BAT glucose uptake. Self-reported endurance athletes had a decrease in cold-stimulated glucose uptake in BAT compared to sedentary male [33] and female subjects [78] after 2 h of cold exposure. These data suggest that in humans, exercise decreases glucose uptake in BAT. The mechanism for this is unclear; while ^18^FDG PET-CT is still considered an important method to identify BAT volume and activity, it is possible the primary role of exercise on BAT is not to stimulate glucose uptake into BAT. It is also probable that exercise-trained subjects have a greater lean mass compared to sedentary subjects and have increased ^18^FDG uptake into their skeletal muscle, causing an underestimation of BAT activity [84]. The effects of exercise on glucose metabolism in BAT have also not been investigated in obese subjects. Thus, more studies are necessary to elucidate the effects of exercise on glucose metabolism in BAT in obese and type 2 diabetic subjects.

Cold exposure and exercise have distinct effects on glucose metabolism in BAT. While cold exposure has been clearly shown to increase BAT glucose uptake [19,22,23], several studies indicate that exercise reduces glucose uptake in BAT [33,73,79] (Figure 1). Nevertheless, during exercise the skeletal muscle requires high levels of glucose uptake and glucose metabolism [80] thus to maintain whole-body homeostasis, the body may decrease glucose uptake into BAT in order to provide substrates to the working skeletal muscle [79,85], and this feature may be shared for certain muscles during mild, acute cold exposure [86]. The differences between cold and exercise-stimulated BAT glucose uptake might be partly related to the increased thermogenic activation of BAT during cold, but not exercise. The beneficial effects of exercise on BAT may be independent of BAT glucose metabolism [15].

## 3. BAT and Fatty Acid Metabolism

Free fatty acids (FFA) are the main substrates for BAT thermogenesis [22]. BAT activation stimulates lipolysis of the intracellular triacylglycerols (TAGs) [22,87] and increases uptake of circulating fatty acids and lipoproteins [88] to provide fuel for thermogenesis in rodents [19,58] and humans [50,77]. Here, we will discuss the role of cold exposure and exercise on fatty acid metabolism in BAT.

### 3.1. Cold Increases Fatty Acid Uptake and Metabolism in BAT

Activation of BAT by cold stimulates the lipolysis of intracellular TAG, releasing long-chain fatty acids that activate UCP1 and increase mitochondrial respiration and thermogenesis in rodents [19,58] and humans [47,77] (Table 1). Cold exposure stimulates the uptake of non-esterified fatty acids (NEFA) from WAT lipolysis and fatty acids from triglyceride-rich lipoproteins (TRL) [19,88] into BAT. Studies in rodents have shown that both acute (1–48 h) [54] and chronic (10 days) [55] cold exposure increase the expression of genes related to FFA synthesis, FFA uptake and oxidation in BAT. These results indicate that cold exposure elicits an increase in FFA uptake and metabolism in BAT. 

Studies in humans have shown that activated BAT is correlated to cold-induced lipolysis, increased FFA re-esterification, FFA oxidation, energy expenditure, and insulin sensitivity compared to individuals with no or minimal BAT activity in both lean [75,76] and obese [50] individuals (Table 2). Additionally, upon mild cold exposure, BAT extracts dietary FA at a higher rate than skeletal muscle [77]. This indicates that BAT is important in lipid and fatty acid metabolism, and that activation of BAT by cold stimulates an increase in fatty acid uptake and metabolism. BAT activation by cold exposure may have beneficial effects on obese and type 2 diabetic patients with altered lipid metabolism. 

### 3.2. Exercise Decreases Lipid Metabolism in BAT

Exercise uses lipids as an energy source for skeletal muscle [89]. In addition to promoting skeletal muscle FFA uptake and utilization, exercise increases FFA mobilization from WAT. With regard to BAT, early studies compared exercise training and cold exposure in rodents and observed that the activity, weight, and lipid content in BAT increased with cold exposure and decreased with exercise [52,53]. Recent studies have provided conflicting evidence regarding the effects of exercise on BAT lipid accumulation and lipolysis (Table 1). One recent study revealed that 3 weeks of wheel-cage exercise in mice increased expression of several genes involved in fatty acid uptake and oxidation in BAT but significantly decreased the pHSL/HSL ratio, indicating a decrease in lipolysis [73], which is consistent with previous reports on the effects of treadmill training on BAT metabolism [70]. Despite the reduction in lipolysis by exercise, the effect on lipid droplet proteins appears to be similar to that of cold-exposed BAT [59] (Table 1). Studies have shown that exercise likely induces lipoprotein remodeling in BAT [70], similar to the remodeling events that occur with cold exposure, and may facilitate increased thermogenesis [59]. Exercise and cold exposure likely prime lipid droplet and FA lipid synthesis, and lipolysis in different ways that are likely to influence thermogenesis and overall activity in BAT. This decrease in lipolysis is the opposite of what occurs during cold exposure. The role of exercise on lipid metabolism in BAT has not been thoroughly investigated; these initial studies indicate that lipid uptake, accumulation, and lipolysis are all increased with cold exposure in rodents and humans, while exercise induces the opposite response (Figure 1). 

## 4. Thermogenic and Mitochondrial Activity in BAT

The thermoregulatory and metabolic actions of BAT are dependent on mitochondrial function [90,91]. In BAT, mitochondria function to dissipate the proton gradient through UCP1 upon activation, thus generating heat [22]. Increased energy intake, a driving force of obesity, may alter mitochondrial function in BAT [92] resulting in impaired thermogenic function. Here, we will discuss the role of cold exposure and exercise training on the mitochondrial activity of BAT. 

### 4.1. Cold Exposure Increases Mitochondrial Content and Activity

Long term cold exposure results in many adaptations in BAT which improve its thermogenic capacity, including increased mitochondrial content and function [22]. Chronic cold exposure in rodents (4 °C–16 °C for 5 to 25 days) increases mitochondrial content, UCP1 expression, and oxidative capacity and respiration [51,59,60,64]. Exposure to cold also increases markers of mitochondrial biogenesis such as PGC1α, NRF1, TFAM, cytochrome c oxidase (COX), and mtDNA [17,29,51,56,62], and oxygen consumption rate [57,63] (Table 1). 

In humans, the mitochondrial enzyme activity in BAT was measured in men who performed work outdoors at very low temperatures compared to workers who performed the same task at ambient temperatures. Outdoor workers had an increase in mitochondrial content and activity of mitochondrial enzymes including β-hydroxybutyrate dehydrogenase, succinate dehydrogenase, and monoamine oxidase [74] (Table 2). It is important to note, however, that a recent study in healthy men demonstrated that the bulk of cold-induced glucose uptake was mediated by deeper and central muscles of the neck and back, as well as the inner thigh muscles [86], and not the BAT. This emphasizes the coordinated metabolic response of BAT and muscle in response to cold. Together, these findings indicate that cold exposure increases mitochondrial content and activity in BAT in both rodents and humans.

### 4.2. Exercise Alters Mitochondrial Content and Activity

Chronic exercise is related to increased mitochondrial activity, biogenesis, and mitophagy in multiple tissues including the heart, skeletal muscle, and white adipose tissue [82,83,93,94,95]. Several studies have examined the effects of exercise on BAT thermogenesis and mitochondrial activity in rodents, with mixed results. Some studies indicated that chronic exercise training increased mitochondrial activity [69], energetics (CPTII, mF1 ATP synthase, MDH) [29], UCP1 content [68], mitochondrial respiration [69,73], and upregulated genes involved in mitochondrial biogenesis in BAT including *Pgc1a*, *Tfam*, and *Nrf1* [29,30,31]. One study showed that, at the ultrastructural level, exercise increased the number of large mitochondria in BAT compared to cold exposure [38]. The functional implications of the larger mitochondria in BAT, or that mechanisms that led to this morphological change, have not been determined (Table 1). Other studies have indicated that exercise does not affect the thermogenic or mitochondrial activity of BAT [53,67], and a third set of studies showed that exercise decreased the thermogenic and mitochondrial activity of BAT [32,33,34]. Recent work in our laboratory showed that 3 weeks of voluntary wheel running in mice decreased functional mitochondrial activity. Basal oxygen consumption rates in the stromal vascular fraction (SVF) isolated from trained BAT differentiated into adipocytes were decreased compared SVF isolated from sedentary BAT. Mitochondrial activity (measured by NADH autofluorescence in vivo) was also significantly decreased after 11 days of exercise [73]. While our data shows that exercise decreases BAT mitochondrial activity, it is possible that different exercise modalities, durations, age of animals, and experimental challenges could induce a different response (Table 1). 

Together these studies suggest that cold exposure increases mitochondrial content and activity in BAT to generate heat production and increase energy expenditure, while the effects of exercise on the mitochondrial activity of BAT are less clear (Figure 1). While this has been investigated in rodents, to our knowledge, this has not been investigated in humans. More work is needed to fully establish the role of exercise on the mitochondrial activity in BAT.

## 5. Cold and Exercise Alter Structural Lipids in BAT

In addition to providing energy, lipids are important central constituents of cellular and organelle membranes [96,97]. The lipidomic profile of BAT is different than that of WAT, likely related to the high density of mitochondria and sympathetic innervation in BAT [98]. Recent studies have investigated the profile of structural lipids in BAT in response to cold and exercise; we will discuss these studies below. 

### 5.1. Cold Induces Species-Specific Changes in BAT Structural Lipids

Recent studies have investigated the effects of cold exposure on the lipidomic profile of BAT using RNA-Seq and mass spectrometry (MS) based lipidomics. Three days of cold exposure (4 °C) resulted in selective remodeling of BAT lipid content, with changes in the fatty acyl composition of TAGs and increased cholesteryl esters (CEs) [61]. The expression of genes related to lipoprotein uptake and turnover (*Lpl*, *Ldlrap1*, and *Lrp5*) and elongation of saturated and monounsaturated C18–C22 fatty acid substrates (*Elovl3*) were also elevated, indicating that cold exposure increases the uptake of fatty acids from lipoproteins into BAT, which are subsequently esterified into TAGs. The composition of TAG species in BAT was also altered in response to cold, with a significant increase in odd-numbered, long- and very-long-chain saturated fatty acyls as a result of increased long-chain fatty acids metabolism, elongation, and esterification into TAGs. 

Three days of cold exposure also induced a selective remodeling of glycerophospholipid species in BAT, with increased 18:0 acyl chains composition in phosphatidylcholine (PC) and phosphatidylethanolamine (PE), 18:0 and 18:1 in lysophosphatidylethanolamines (LPE), and 18:2 in phosphatidylserine (PS). The increase in 18:0 acyl chains leads to increased mitochondrial respiratory capacity after cold exposure [99]. Additionally, genes involved in synthesis and remodeling of glycerophospholipids in BAT were also increased after cold exposure. Glycerophospholipids are important components of cellular and mitochondrial membranes [100] and can act in important signaling pathways related to brown adipocyte proliferation, lipid oxidation, and thermogenesis through the activation of ligand-activated transcription factors. Thus, the cold-mediated changes in glycerophospholipid subspecies may affect signaling cascades and transcription factors in the cold-induced thermogenic adaptation. 

Similarly, another study investigated the effects of 7 days of cold exposure (5 °C) on the lipidomic profile of serum and BAT [65] (Table 1). Seven days of cold exposure resulted in selective remodeling of glycerophospholipids in BAT, increasing phosphatidylglycerol (PG) and cardiolipins (CL) species. The increased CL levels were also observed in the serum of humans exposed to 14 °C for 1 h. PG is a phospholipid precursor for CL [101], and CL constitutes up to 20% of total mitochondrial membrane lipids. CL binds directly to UCP1 to increase its tethering within the mitochondrial membrane [102]. Another study investigating the effects of chronic cold exposure (3 weeks) also determined an increase in PG and CL in BAT [66], and an increase in the enzyme CL synthase 1 (CRLS1), which was correlated with the increases in UCP1 expression, uncoupled respiration, glucose uptake, and systemic insulin sensitivity. Since CL production is reduced in BAT in obese mice [103] and alterations in CL have been associated with mitochondrial dysfunction in several pathological conditions and diseases including obesity and T2DM [103,104,105], a cold-induced increase in CLs in BAT supports a potential mechanism through which cold exposure could help control systemic glucose regulation. Taken together, these studies show that changes in structural lipid metabolism by cold exposure can exert an important function on BAT thermogenesis and improve metabolic health, and these adaptations to structural lipids are observed as early as 3 days after cold exposure.

### 5.2. Exercise Induced Changes to BAT Structural Lipids Are the Opposite of Cold Exposure

A recent study from our laboratory investigated the effects of exercise on the structural lipid profile of BAT using MS/MS^ALL^ lipidomics [71]. After 3 weeks of voluntary wheel-cage running, the overall abundance of TAGs was significantly reduced in BAT after exercise, an effect that was not observed after cold exposure [61]. In fact, in contrast to cold exposure, exercise reduced the expression of genes involved in fatty acid biosynthesis (*Acaca*, *Scd1*, *Agpat3*, *Dgkd*, and *Mlxipl*). After cold exposure, BAT increases TAG production in order to use energy from the TAGs as heat and contribute to non-shivering thermogenesis. Conversely, thermogenesis in BAT during or immediately after exercise remains unchanged, thus the de novo synthesis of FFA and TAG would not be necessary to sustain BAT activity. 

Consistently, the phospholipid pathway appears to be regulated differently in response to exercise and cold exposure in BAT. Exercise increases specific acyl chains 36:2 in phosphatidylcholines (PC), 40:5 and 40:6 acyl chains in phosphatidylethanolamine (PE), and 16:0 and 16:1 acyl chains in phosphatidylserine (PS) species [61]. The molecular species that are increased after exercise are distinct from those upregulated after cold exposure (18:0, 18:1, and 18:2 acyl chains) [61]. The expression of genes involved in phospholipid metabolism (*Agpat3*, *Gpd1*, *Lgpat1*, *Ptdss2*, and *Pld1*) are also significantly decreased in BAT after exercise. Interestingly, and in contrast to chronic cold exposure [65,66] (Table 1), the overall abundance of lysophosphatidylglycerols (LPG) and CL were decreased after chronic exercise. Despite the extensive characterization of the lipidomic changes in response to both cold and exercise, to our knowledge no studies have investigated the role of the metabolic differences and specific molecular targets that regulate the remodeling of structural lipids [106]. Taken together, these studies indicate that cold and exercise produce profound species-specific modifications to the BAT lipidome that might confer enhanced thermogenic or endocrine capabilities to BAT. Previous studies have hypothesized that the cold-induced lipidomic changes in BAT might be necessary for mitochondrial biogenesis and enhancement of mitochondrial function, independently from the thermogenic activation [65]. Furthermore, the lipidomic changes elicited by exercise training might prime not only BAT [71] but also skeletal muscle mitochondria [106]. Thus, the lipidomic signature in response to cold and exercise are largely required to sustain such metabolic and energetic adaptations. Despite this information, the mechanistic importance of these differences is still unclear and the physiological effects of cold- and exercise-induced changes in the BAT lipidome remain important topics of investigation (Figure 1).

## 6. “Batokines”: Secreted Factors Released by Cold and Exercise

Adipokines released from BAT, or “batokines,” can act locally or systemically to improve metabolic health [20]. In this section we will review several batokines that are affected by cold, exercise, or both.

### 6.1. 12,13-diHOME Increases FA Uptake in BAT and Skeletal Muscle

Circulating lipids have been identified to be released from certain tissues and act locally or systemically to promote glucose tolerance and insulin sensitivity. A recent study investigated the effects of cold on signaling lipids in humans and rodents [107]. Using mass-spectrometry based lipidomics, the lipokine 12,13-diHOME was identified to be released from BAT after 1 h of cold exposure in both rodents (4 °C) and humans (14 °C). Expression of epoxide hydrolase 1 and epoxide hydrolase 2 (*Ephx1* and *Ephx2*), the enzymes involved in 12,13-diHOME synthesis from linoleic acid, were significantly increased in BAT in response to cold exposure. The release of 12,13-diHOME increased fatty acid uptake, lipolysis, and thermogenesis in BAT. In addition, prolonged treatment with 12,13-diHOME in obese mice decreased circulating triglyceride levels. This is the first study to identify a signaling lipid released from BAT in response to cold that functions in an autocrine-paracrine mechanism to improve metabolic health. 

A recent study in our lab investigated the effects of exercise on circulating lipokines in rodents and humans [27]. Surprisingly, we determined that acute physical exercise in humans and mice increased the circulating levels of the lipokine 12,13-diHOME. When BAT was surgically removed from mice and the mice underwent an acute bout of exercise, there was no increase in 12,13-diHOME, indicating that BAT was the source of this lipokine. 12,13-diHOME was positively correlated with VO_2_ peak and negatively correlated with BMI. Furthermore, acute treatment with 12,13-diHOME in mice in vivo increased skeletal muscle fatty acid uptake and oxidation but had no effect on glucose uptake. These data are the first to identify an endocrine role for BAT in response to exercise and indicate a novel mechanism for BAT-skeletal muscle cross-talk. 

The parallels between the effects of exercise and cold exposure on 12,13-diHOME are important and somewhat unexpected [27]. First, both short term and chronic cold exposure or exercise increase the concentration of circulating 12,13-diHOME to a similar extent. Both cold exposure and exercise increase 12,13-diHOME in BAT, which is the tissue source of circulating 12,13-diHOME. This is unexpected because while cold exposure is a well-established tool to stimulate BAT activity, most investigations have shown that exercise training has an opposite effect than cold exposure and decreases BAT activity in humans and rodents [32,33]. It is possible that that cold exposure causes the release of 12,13-diHOME from BAT to act in an autocrine manner and provide fuel for the BAT, while exercise causes the release of 12,13-diHOME from BAT to act in an endocrine manner, stimulating uptake of fatty acids into the working skeletal muscle [27]. Another potential hypothesis is that exercise increases the sympathetic tone and vascularization of BAT, similar to cold exposure, despite the weak activation of thermogenesis [38]. This could allow for a similar effect of exercise and cold exposure on 12,13-diHOME (Figure 1).

### 6.2. Fibroblast Growth Factor 21 (FGF21) Regulates Glucose and Lipid Metabolism

One of the most well-investigated batokines is FGF21, a peptide hormone that is involved in both lipid and glucose metabolism [108]. The predominant source of FGF21 is the liver, but FGF21 is also highly expressed in WAT [109] and BAT [110]. FGF21 knockout (KO) mice show larger lipid droplets in BAT and cold intolerance [111]. Liver-derived FGF21 can activate thermogenesis in neonatal BAT [112], and treatment with FGF21 in mice increases expression of UCP1 [113], improves insulin sensitivity, and lowers blood glucose and lipids [108]. FGF21 effects in BAT include acutely enhancing in insulin sensitivity, glucose uptake, and accelerated lipoprotein catabolism [110,114,115]. 

Short-term cold exposure in rodents increases the expression and release of FGF21 from BAT [108,110] via a cAMP-related mechanism. Infusion of FGF21 in mice increases energy expenditure and core temperature, likely indicating catabolism of glucose and fatty acids and possibly thermogenic activation of BAT [113,116,117]. A study comparing wild type (WT) mice with FGF21 KO also demonstrated that cold exposure (18 °C to 5 °C for 5 weeks) increases FGF21 gene expression in BAT of WT but FGF21 is not required for BAT cold adaptation in FGF21 KO [118]. In humans, FGF21 is highly expressed in BAT [119]. Mild cold exposure (19 °C) increases circulating FGF1 levels [120], and cold induced circulating FGF21 levels are associated with increased BAT activity [121,122]. A clinical model of cold exposure (12 min from 18 °C to 12 °C) in healthy human subjects increased plasma FGF21, but the source of the circulating FGF21 was not determined [122]. Thus, cold exposure can increase FGF21 expression in BAT of rodents and humans and it is possibly associated with increased BAT function and increased glucose and fatty acids utilization; however, FGF21 is not essential for the maintenance of metabolic homeostasis in the cold adaptation [118]. 

Acute exercise increased FGF21 in the serum and in liver, but not in white adipose tissue of rodents [123], while the role of BAT-released FGF21 in response to exercise was not established. Acute [123] and chronic exercise also increased serum FGF21 in healthy human subjects [124], and the liver was indicated as the primary source of FGF21 in response to exercise [123]. The effects of exercise in BAT FGF21 expression and release was not yet determined. While there is no evidence comparing BAT-derived FGF21 to recombinant FGF21 administration, a long-acting analog of FGF21 (PF-05231023) has demonstrated to be effective at reducing body weight and serum TAG, while simultaneously increasing the anti-inflammatory adipokine adiponectin in non-human primates and humans [125]. Additionally, we have demonstrated that BAT transplantation leads to improved glucose tolerance, increased insulin sensitivity, reduced body weight and fat mass, and reversal of high fat-diet (HFD)-induced insulin resistance in mice [20]. To our knowledge, FGF21 isolated from BAT has not been compared to recombinant FGF21 in mice or humans. Future studies should compare the cold- and exercise-induced FGF21 with a synthetic FGF21 to assess the translatability of these therapeutic strategies. Taken together, these data show that FGF21 can be released from BAT and plays an important role in glucose and lipid metabolism. FGF21 is regulated by both cold and exercise, and while cold increases FGF21 release from BAT in rodents, it is unclear whether the tissue responsible for the increase in response to exercise is BAT; this will be the topic of future investigations (Figure 1).

### 6.3. Vascular Endothelial Growth Factor A (VEGFA) Mediates Thermogenic Adaptations

VEGFA is an angiogenic growth factor that stimulates vascular endothelial cell activation, proliferation, and migration [126]. VEGFA is also a batokine that acts in a paracrine fashion to regulate vascularization [56] and activate thermogenesis. Studies in both rodents [56] and humans [45] demonstrate a direct relationship between increased VEGFA, vascularization in BAT, and increased thermogenesis. 

Cold exposure studies utilizing transgenic mouse models have clearly illustrated the importance of VEGFA in thermogenesis. Overexpression of VEGFA increases vascularization of BAT and expression of both *Ucp1* and *Pgc1α* [57], increasing thermogenic capacity upon cold exposure. Alternatively, VEGFA-null mice display functional decrements in regards to thermogenesis due to reduced vascularization and mitochondrial function in BAT [127].

To the best of our knowledge, studies have not looked at the effects of exercise on VEGFA specific to BAT. However, VEGFA is increased in WAT in response to exercise in both mice [128,129] and humans [130]. Because VEGFA is a crucial protein that regulates adaptations to exercise training and improvements in aerobic capacity [131], it is possible that exercise has a similar affect to increase VEGFA in BAT. It is tempting to hypothesize that adaptations to VEGFA content in response to exercise may be related to BAT activity, but this has not yet been investigated; this will be an important focus of future investigations (Figure 1).

## 7. Cold and Exercise Induce a “Beiging” of White Adipose Tissue (WAT)

In addition to the stimulation of BAT, induction of brown-like or “beige” cells by cold and exercise has gained notoriety given its high therapeutic potential [132,133,134]. Since the discovery of cold-induced BAT in adult humans, researchers have studied biopsies of ^18^FDG-positive neck fat depots to establish the genetic, molecular, and functional signature of human BAT to understand whether these adipocytes recapitulate aspects of brown or brown-like adipose depots—the so called beige adipose. Beige fat cells originate from WAT and share many characteristics with brown adipocytes, such as a multilocular lipid droplets, increased mitochondrial density, and high levels of UCP1, together with increased capacity for fuel oxidation and thermogenesis [135,136,137,138]. Currently, there is evidence that human BAT displays either a classical brown or a beige signature, or a combination of both, which explains the heterogeneous responses [138,139,140,141,142,143]. Interestingly, both cold and exercise are important inducers of the beige program or “beiging” of white adipocytes [28,144]. Cold- and exercise-induced beiging increases the number of metabolically active cells within WAT depots, which in turn constitutes an effective strategy to combat obesity and T2DM [132,133,134]. Increase in the number and the activity of beige cells, as well as the reprogramming of cellular beige precursors might be responsible, at least in part, for the beneficial effects of cold and exercise on metabolism. Here, we will summarize the role of cold exposure and exercise training on adipose tissue beiging. 

### 7.1. Long-Term Cold Exposure Induces Beiging of WAT

Chronic cold exposure induces sympathetic activation and thermogenic response in WAT. The interaction of norepinephrine with β3-ARs present on the membrane of white adipocytes initiates a cascade that leads to the overexpression of UCP1 and other thermogenic proteins [137,145]. Recent studies have demonstrated that cold exposure could also stimulate beiging of WAT through the batokine signaling such as interleukin 6 (IL-6), neuregulin 4 (Nrg4), and FGF21 [146,147,148], and the myokine irisin [149]; these browning agents have been previously reviewed [150,151,152]. Studies in rodents demonstrate that cold adaptation (4 °C to 6 °C) results in beiging of subcutaneous WAT (scWAT) with improved metabolic activity [153,154,155,156,157].

Although experiments in rodents have shown a profound effect of cold on beiging, experimental evidence in humans remains inconclusive. Studies with cultured healthy human scWAT cells demonstrated that long-term cold exposure increases PGC1α, UCP1 expression, and mitochondrial activity, which are all defining features of beige cells [158,159]. In spite of this, a study of cold acclimation in healthy humans (15 °C–16 °C, 2 h to 6 h/day, 10 days) did not show beiging of scWAT [160]. These disparities in the results are likely associated with the duration of cold exposure. While it seems that long term cold exposure can promote beiging of scWAT in humans, further studies are needed to define the length of cold exposure needed to induce beiging in humans, and whether these observations can be translated to subjects with obesity and T2DM.

### 7.2. Effects of Exercise Training on the Beiging of WAT

In recent years, studies in rodents have extensively demonstrated that chronic exercise, through various mechanisms including sympathetic activation [145,161] and secreted factors myokines, induces a beiging of WAT (irisin, IL-6, and meteorin-like (Metrnl)) [162,163,164,165,166]. This exercise-induced beiging is characterized by increased markers such as Ucp1, Pgc1α, Prdm16, and Cidea in lean rodents [73,129,162,167]. In HFD-induced obese animals, chronic exercise shows conflicting results on beiging. Some studies indicate that exercise increased beiging markers [32,168], whereas another found a neutral effect [169], and some observed decreased beiging markers following chronic exercise training [170]. These contradictory results are likely explained by differences in the modality and duration of exercise, which warrants further investigation. 

The exercise-induced beiging in humans remains largely undefined [33,171]. Studies assessing the effect of chronic exercise on beiging did not observe such effects in scWAT biopsies in healthy [33,171,172] or obese individuals [173,174]. A recent study however, demonstrated a significant effect of exercise training on beiging markers (Ucp1 and Cpt1) in abdominal scWAT across weight spectrums, despite the lack of effects on insulin sensitivity [175]. Additional studies are necessary to assess different exercise types, duration, frequency, and intensity, as well as gender, age, and health status, and standardization of adipose tissue sampled, and methods used to assess beiging of WAT. 

Taken together, both cold and exercise can induce beiging in lean rodents leading to increases in metabolically active adipose tissue. In humans, however, the clinical evidence remains insufficient to suggest that both stimuli have the positive effects on WAT beiging.

## 8. Conclusions and Future Perspectives

Over the last decade BAT has garnered great interest as a therapeutic target to combat obesity and T2DM as numerous studies have established an association between BAT activity and metabolic health. Both cold exposure and exercise represent potential tools to impact BAT function by altering its activity and metabolism. Cold exposure is an effective method to activate the thermogenic activity of BAT, increase energy expenditure in BAT, and improve glucose homeostasis, insulin sensitivity, and lipid metabolism. Exercise is an important therapeutic tool to regulate whole-body glucose homeostasis, but the role of exercise on glucose uptake, lipid metabolism, and mitochondrial activity in BAT is unclear. There are several potential reasons for the conflicting effects of exercise on BAT activation, including differences in modality and duration of exercise, strain of mouse used during the investigation, and different measurements of BAT activation. Thus, greater standardization of testing protocols is needed to make true comparisons and interpretations of the data.

Because of the distinct effects of cold and exercise on BAT metabolism, there is a great need to investigate the synergistic effects of a combined model of cold and exercise. An important overlap exists between the effects of cold and exercise on multiple batokines, which emphasizes the need for the investigation of the synergistic effects of cold and exercise on batokine release. Because both cold exposure and exercise can stimulate the release of batokines that leads to glucose and fatty acid metabolism improvement (i.e., FGF21 and 12,13-diHOME), this new area of research will likely continue to drive interest in the field of BAT metabolism, thereby increasing our knowledge and ability to harness the full potential of BAT as a therapeutic tool to ameliorate metabolic diseases, specifically T2DM. Interestingly, endurance athletes have begun to combine extreme but acute (≈3 min) bouts of cold exposure called “cryotherapy” with their training schedules as a means of speeding recovery [176]. This could be a potential therapeutic tool to combat metabolic abnormalities, for instance, targeting BAT activation in type 2 diabetic patient populations leading to concomitant improvements in insulin sensitivity. It is important to note that some obese patients, especially cases of morbid obesity, are related to the difficulty in performing physical exercises, considering the respiratory difficulty and joint diseases associated with excess body mass [177,178]. In addition, cold exposure protocols can be quite unsuitable and uncomfortable for obese patients [137]. Thus, these limitations need to be taken into consideration for the elaboration of a therapeutic protocol associating exercise and cold exposure in obese individuals.

Future studies will fully elucidate the mechanisms through which these treatments cause adaptations to BAT and affect metabolic health. As both therapies can improve insulin sensitivity, a combined treatment may be an attractive strategy to employ comprehensive lifestyle modifications for obese patients with a high risk of developing T2DM and other metabolic derangements. 

## Figures and Tables

**Figure 1 biology-08-00009-f001:**
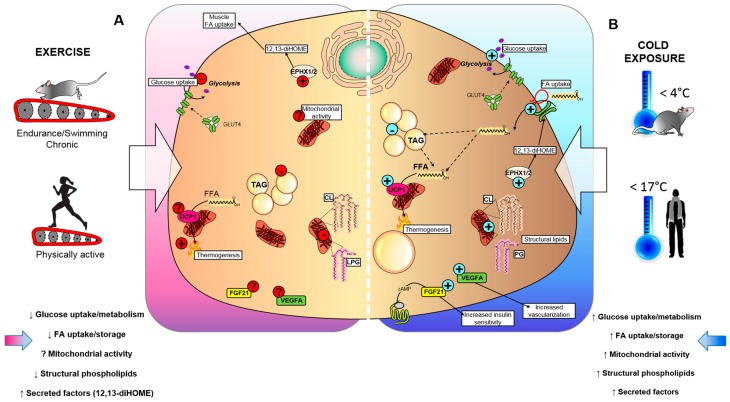
Effects of cold exposure and exercise on BAT. (**A**) Exercise and (**B**) cold exposure effects on BAT metabolism can cause the release of batokines, which act in an autocrine, paracrine, or endocrine manner to influence metabolic health. (**A**) Exercise reduces insulin-stimulated glucose uptake in BAT, suppresses triglyceride (TAG) accumulation, and lowers mitochondrial lipids, such as cardiolipin (CL) and lysophosphatidylglycerol (LPG), which could affect the thermogenic capacity of BAT. Conversely, exercise training stimulates epoxide hydrolase 1 and 2 (Ephx1/2), and increases the synthesis of the lipokine 12,13-diHOME. The effects of exercise on mitochondrial activity, fibroblast growth factor 21 (FGF21) and vascular endothelial growth factor A (VEGFA) production are unknown. (**B**) Exposure to cold temperatures stimulates insulin-stimulated glucose uptake in BAT, synthesis of CL and LPG, and secretion of batokines FGF21 and VEGFA that lead to increased insulin sensitivity and tissue vascularization. Additionally, cold exposure increases the synthesis of 12,13-diHOME, which can act in an autocrine manner to promote fatty acid uptake in BAT, ultimately leading to increased TAG and UCP1-mediated thermogenesis.

**Table 1 biology-08-00009-t001:** Effect of cold and exercise on BAT metabolism in mice.

Therapeutic Strategy	Reference	Treatment	Duration	Animal Model	Experimental Challenge	BAT Glucose Metabolism	BAT Lipid Metabolism	BAT Mitochondrial Activity
**Cold exposure**	[51]	5 °C	4 weeks	Rats	NA	NA	NA	↑ Mitochondrial enzyme activity: cytochrome c; palmitoyl-CoA oxidase; catalase; acid phosphatase and peroxisomal activity
**Cold exposure**	[52]	−5 °C	2 h/day 4 weeks	Rats	NE-induced thermogenesis	NA	↑ BAT weight and lipid content	NA
**Cold exposure**	[53]	−15 °C	2 h/day for 10 weeks	Rats		NA	↑ BAT weight	NA
**Cold exposure**	[41]	5 °C	48 h	Rats	48 h fasting	↑ 2-[3H]deoxyglucose uptake	NA	NA
**Cold exposure**	[42]	4 °C	Acute: 4 h Chronic: 1–10 days	Female Rats	24 h fasting	Acute: ↑ 2-deoxyglucose uptake Chronic: ↑ 2-deoxyglucose uptake ↑ GLUT4 expression	NA	NA
**Cold exposure**	[54]	4 °C	1 to 48 h	Mice	NA	↑ Gene expression: PDH; PFK-C; PFK-1; GLUT4; LDH ↓ Gene expression: GLUT1; GLUT2; GLUT3 ↑2-Deoxyglucose uptake	↑ Gene expression: ATP-citrate lyase; FAS; GPAT; ACC1; ACC2; LPL; MG lipase; LCAD; MCAD ↑ LCFA carboxyl-carbon into BAT (FA uptake) ↑ LCFA carboxyl in the acid-soluble products (FA utilization)	NA
**Cold exposure**	[55]	4 °C	10 days	Rats	NA	NA	↑ Glyceroneogenesis ↑ Glucose incorporation to glycerol ↑ LPL activity	NA
**Cold exposure**	[40]	4 °C	4 h	Female Rats	NA	↑ ^18^F- or (3)H-FDG uptake ↑ GLUT1 imunohistolocalization	↓ Lipid droplet size	↑ UCP1 immunohistolocalization
**Cold exposure**	[56]	4 °C	5 weeks	Mice	NA	NA	↓ Adipocyte size	↑ Mitochondria number ↑ COX4 expression.
**Cold exposure**	[19]	4 °C	24 h	Mice	NA	↑ ^14^C-desoxyglucose uptake ↑ Gene expression: GLUT1 and GLUT4 ↑ Protein expression: P-S6 (down-stream target of the insulin-AKT-signaling cascade)	↑ BAT uptake of TRL ↑ LPL activity ↑ CD36 expression	NA
**Cold exposure**	[57]	4 °C	Acute and 3 weeks	UCP1-rtTA mice	Overexpression of VEGF-A	NA	NA	↑ Mitochondrial ↑ Oxygen consumption rate
**Cold exposure**	[58]	4 °C	7, 14, and 21 days	Mice	NA	NA	↓ Lipid droplet size	NA
**Cold exposure**	[59]	4 °C	4 h, 24 h, 48 h, and 72 h	Mice	NA	NA	↑ Protein expression: PLIN1; PLIN2/ADRP; ATGL; HSL; pHSL ↑ Gene expression: ADRP	↑ Protein expression: UCP1; CIDEA ↑ Gene expression: UCP1; PGC1α ↑ Mitochondrial cristae biogenesis
**Cold exposure**	[60]	4 °C	0.5 to 10 days	AXB8 mice	NA	NA	NA	↑ Protein expression: UCP1; COXIV; CYTC ↑ Gene expression: UCP1
**Cold exposure**	[61]	4 °C	3 days	Mice	NA	NA	↑ Gene expression: LPL; LDLrap1; LRP5; Elovl3 ↑ Cholesteryl Esters	NA
**Cold exposure**	[62]	4 °C	1 to 7 days	Mice	NA	↑GLUT1 gene expression (only days 1 and 2)	↑ Protein expression: PPARγ; aP2 ↑ Gene expression: Elovl3 ↓ Lipid droplet size	↑ Protein expression: UCP1 ↑ Gene expression: PGC1α; UCP1
**Cold exposure**	[63]	4 °C	24 h, 48 h, 96 h	Them1−/− and Them1+/+ mice	NA	NA	↓ Triglyceride content	↑ UCP1 Protein expression (96 h) ↑ Oxygen Consumption Rate
**Cold exposure**	[64]	16 °C and 4 °C	16 °C for 2 weeks and 4 °C for more than 2 weeks	WT, UCP1−/−, and SLN−/− mice	NA	NA	↓ Lipid droplet size (WT and SLN−/−)	↑ UCP1 imunohistolocalization ↑ protein expression: UCP1; TFAM, complex1 and 2 ↑ Mitochondria cristae (results found in WT and SLN−/−)
**Cold exposure**	[65]	5 °C	7 days	Mice	NA	NA	Remodeling of glycerophospholipids	↑ Cardiolipins
**Cold exposure**	[66]	5 °C	3 h, 3 days, or 3 weeks	Mice	NA	↑ Glucose metabolism signaling pathway	↑ Signaling pathways: phospholipids metabolism; TAG biosynthesis; Glycerophospholipid biosynthesis; Fatty Acyl-CoA biosynthesis.	↑ CRLS1 expression ↑ Cardiolipins ↑ TCA pathway ↑ ETC pathway
**Exercise**	[52]	Treadmill training (25 m/min)	4 weeks	Rats	Warm (24 °C) and cold acclimation (−5 °C, 2 h/day); All groups: NE-induced thermogenesis	NA	↓ Lipid content in warm acclimated trained compared to warm-acclimated sedentary. No differences in lipid content in cold-acclimated sedentary versus trained.	NA
**Exercise**	[67]	Treadmill training	90 min/day 6 weeks	Male rats	NE stimulation	NA	NA	↓ BAT Blood flow (but blood flow does not change between groups under NE stimulation)
**Exercise**	[53]	Swimming	2 h/day for 10 weeks	Rats	NA	NA	↓ BAT weight	NA
**Exercise**	[31]	Treadmill training (23 m/min)	6 weeks	Ovariectomized rats	NA	NA	NA	↑ Mitochondrial protein content ↑ Cytochrome oxidase activity ↑ Oxygen consumption
**Exercise**	[68]	Swimming	6 weeks	Male young and old mice	NA	NA	↑ BAT weight	↑ Mitochondrial protein content ↑ UCP1 antigen level ↑ GDP binding (indicator of UCP1 activity)
**Exercise**	[34]	Swimming	21 days	Rats	TSH-stimulation test in cold (4 °C) for 30 min or no TST in 30 °C for 30 min	NA	NA	↓ 5′ deiodinating activity
**Exercise**	[30]	Treadmill training (15 m/min)	8 weeks	Chow diet mice and HFD mice	NA	NA	NA	↑ UCP1 and Dio2 expression
**Exercise**	[69]	Swimming	8 weeks	Ovariectomized rats	NA	NA	NA	↑ Mitochondrial oxygen consumption
**Exercise**	[38]	Endurance (≈60% of VO_2_max), 5 days/week	1 and 6 weeks.	Male Sprague Dawley rats	With or without recovery	No change in GLUT1 and 4 expression	↑ Unilocular lipid droplet	↑ PGC-1α and PGC-1β expression and weak UCP1 ↑ parenchymal vascularization ↑ MCT-1 lactate transporter
**Exercise**	[32]	Treadmill training (70–85% VO_2_max	8 weeks	Rats	NA	NA	↑ Lipid droplet area.	↓ Protein expression: UCP1; PGC1α ↓ Palmitate oxidation.
**Exercise**	[70]	Progressive treadmill training (18–25 m/min for 30–60 min at 10% incline)	8 weeks	Male rats	NA	NA	↑ Storage protein PLIN5 No change in synthesis FAS but ↓ in ACC ↓ pHSL/HSL protein expression	No changes in mitochondrial proteins COX IV, PDH, UCP1
**Exercise**	[71]	Wheel cage running	3 weeks	Male mice	NA	NA	↓ Fatty acid biosynthesis gene expression: Acaca; Scd1; Agpat3; Dgkd; Mlxipl ↓ Overall abundance of TAG	NA
**Exercise**	[29]	Treadmill training, 55–65% of maximal running speed (15–20 m/min)	8 weeks	Rats	NA	↑ Insulin signaling protein expression: IR; p-IRS-1; pERK	NA	↑ Protein expression: NRF1; TFAM; PGC1α; SIRT1; pAMPK/AMPK ratio; ATP synthase; mMDH; UCP1; UCP2; UCP3
**Exercise**	[72]	Swimming in low and moderate intensities	13 weeks	Metabolic syndrome, high fat fed (30% lard) rats	NA	↑ Akt-2 and GLUT4 gene expression	NA	NA
**Exercise**	[73]	Wheel cage running	3 weeks	Mice	NA	↑ Gene expression: GLUT4; Hk2; Eno1 ↓ Glucose uptake	↑ Gene expression: Fabp3; Acsl3; Gpd1; Gyk ↓ pHSL/HSL protein expression	↑ Gene expression: Cidea; Cd36; Citrate synthase; UCP1 ↓ NADH fluorescence intensity ↓ Oxygen consumption rate

NA: Not available; PDH: Pyruvate dehydrogenase; PFK: Phosphofructokinase; LDH: Lactate dehydrogenase; FAS: Fatty Acid Synthase; GPAT: Glycerol-3-Phosphate Acyltransferase; ACC: Acetyl-CoA carboxylase; LPL: lipoprotein lipase; UCP1: mitochondrial uncoupling protein 1; MG: monoacylglycerol; LACD: Long-chain acyl-CoA dehydrogenase; MCAD: medium-chain acyl-CoA dehydrogenase; LCFA: Long-chain Fatty Acids; COX: Cytochrome c oxidase; AKT: RAC-alpha serine/threonine-protein kinase (PKB); CD36: cluster of differentiation 36; VEGF: Vascular endothelial growth factor; PLIN: Perilipin; ADRP: Adipose differentiation-related protein; ATGL: Adipose triglyceride lipase; HSL: Hormone-sensitive lipase; CIDEA: Cide domain-containing protein Cidea; PGC1α: Peroxisome proliferator-activated receptor gamma coactivator 1-alpha; CYTC: Cytochrome C; LDL: low-density lipoproteins; LRP: Lipoprotein receptor-related protein; ELOVL: Elongation of very long chain fatty acids protein; PPARγ: peroxisome proliferator-activated receptors gamma; WT: Wild Type; SLN: Sarcolipin; TFAM: Mitochondrial transcription factor A; CRLS: Cardiolipin Synthase; TCA: tricarboxylic acid cycle; ETC: electron transport chain; GDP: guanine diphosphate; TSH: Thyroid-stimulating hormone; HFD: High Fat Diet; Dio2: Type II iodothyronine deiodinase; MCT-1: Proton-linked monocarboxylate transporter, member 1; Acaca: Acetyl-CoA Carboxylase Alpha; Scd1: stearoyl-CoA desaturase 1; AGPAT: Acyl-CoA:glycerol-3-phosphate acyltransferase; Dgkd: Diacylglycerol kinase delta; Mlxipl: MLX-interacting protein-like; IR: Insulin Receptor; IRS: Insulin Receptor Substrate; ERK: extracellular signal–regulated kinase; NRF: Nuclear respiratory factor; SIRT1: Sirtuin 1; mMDH: Mitochondrial malate dehydrogenase; Hk2: Hexokinase 2; Eno1: Enolase 1; Fabp3: fatty acid-binding protein 3; Acsl3: Long-chain-fatty-acid—CoA ligase 3; Gpd1: Glycerol-3-phosphate dehydrogenase; Gyk: glycerokinase; NADH: Reduced Nicotinamide adenine dinucleotide.

**Table 2 biology-08-00009-t002:** Effect of cold and exercise on BAT metabolism in humans.

Therapeutic Strategy	Reference	Treatment	Duration	Study Subjects	Experimental Challenge	Effects on Glucose Metabolism	Effects on Lipid Metabolism	Effects on BAT Mitochondrial Activity
**Cold exposure**	[74]	Cold outdoor weather in Northern Finland	NA	Male and female outdoor workers	NA	NA	NA	↑ Enzyme activity: β-hydroxybutyrate dehydrogenase; Succinate dehydrogenase; Monoamine oxidase
**Cold exposure**	[46]	16 °C	2 h	Male	NA	↑ ^18^F-FDG uptake	NA	NA
**Cold exposure**	[45]	17 °C	2 h	Male and female	Foot in cold water (5 min in/5 min out PET/CT session	↑ ^18^F-FDG uptake	NA	NA
**Cold exposure**	[44]	19 °C room and feet on an ice block intermittently (4 min every 5 min)	1 h	Male and female	NA	↑ ^18^F-FDG uptake	NA	NA
**Cold exposure**	[75]	19 °C and were decreased by 1 °C approximately every 30 min until shivering.	5−8 h	Male	NA	↑ ^18^F-FDG uptake	NA	↑UCP1 imunohistolocalization; ↑ Uncoupled Mitochondrial respiration
**Cold exposure**	[48]	14 °C−15 °C	10 days	Male T2DM subjects	NA	↑ ^18^F-FDG uptake	NA	NA
**Cold exposure**	[76]	18 °C	2 h	Male T2DM	NA	NA	↑ ^18^FTHA uptake	NA
**Cold exposure**	[47]	19 °C and were decreased by 1 °C approximately every 30 min until subjects reported shivering	6 h	Male	NA	↑ ^18^F-FDG uptake	NA	NA
**Cold exposure**	[50]	18 °C	5−8 h	Male obese	NA	↑ ^18^F-FDG uptake	↑ LPL, CD36 gene expression. Association between BAT volume and FFA uptake and oxidation	↑ UCP1gene expression ↑ Oxygen consumption rate
**Cold exposure**	[77]	18°C	5 h and 4 weeks	Male	NA	NA	↑ DFA and ^18^FTHA uptake in acute stimulation; no further increase after chronic stimulation	NA
**Exercise**	[33]	Endurance-trained athletes	NA	Males	2 h cold exposure	↓ [^18^F]FDG uptake	NA	NA
**Exercise**	[78]	Athletes versus non-athletes	NA	Female	14 °C for 120 min	Trend to ↓ [^18^F]FDG uptake	NA	NA
**Exercise**	[79]	HIIT and moderate continuous training	2 weeks	Healthy middle-aged men	Insulin stimulated glucose uptake	No changes in glucose uptake	NA	NA

NA: Not available; LPL: lipoprotein lipase; MG: monoacylglycerol; CD36: cluster of differentiation 36; UCP1: mitochondrial uncoupling protein 1; ^18^FTHA: 14(R, S)-[(18)F]Fluoro-6-thia-heptadecanoic acid; DFA: dietary fatty acids; HIIT: High-intensity interval training.

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
