# Peer review of "Cold and Exercise: Therapeutic Tools to Activate Brown Adipose Tissue and Combat Obesity"

_biology, 2019, doi:10.3390/biology8010009_

Round 1
Reviewer 1 Report
This review by da Silva et al. aims to compare and contrast exercise and cold exposure as potential therapeutic modalities to stimulate BAT as an anti-obesity treatment. The inclusion of new data regarding lipidomics in BAT with these two interventions make the review timely and novel, however numerous inaccuracies are included in regards to other better studied aspects of BAT function and metabolism.
Itemized Edits:
Fig. 1 legend is not adequately describing the Figure - what is going on inside the cell? Descriptions of cold vs exercise are a bit misleading - for example, if FA uptake/storage in BAT goes down with exercise then why does line 62 say circulating lipids are down? Are these not connected - this needs to be made clear. FA oxidation should be mentioned in comparison of the two interventions. Specific structural lipids should be listed. See comments below as well.
Line 45 - sensory nerves are also present
line 52, should say 'that influence' not 'for influence'
don't under line insulin sensitivity and glucose tolerance in line 61
Add detail to descriptions of previous research findings - for example in line 65-66: did this depend on intervention? mouse vs human? depot of BAT under study (remember - there are many in mouse AND human!)? Also remember that most human BAT is more like the inducible/beige cells in mouse...this should be made clear and considered as part of the assessments of the research literature
Nerves are clearly important for the processes reviewed but are only mentioned cursorily. The possibility that increased sympathetic tone in the body after exercise could have off-target effects on BAT stimulation should be mentioned (see the numerous commentary articles on this topic, especially around the discovery of irisin, which is for some reason completely ignored in this review)
no comma after Activated (line 84)
all data being referred to should be in the past tense - for example, line 90 should say: increased, improved, etc.
how was insulin sensitivity measured (line 90-91) - this level of detail should be improved throughout so that the reader has a sense of what was done in the studies being referenced
is there not any data on cold exposure affecting BAT glucose uptake directly? That is hard to believe...
which obese, glucose-intolerant mice are referenced in line 93? db/db? This type of detail matters in interpreting the data
insulin sensitivity is mentioned in line 96 but no data are described on this topic - please expand. Insulin sensitivity is different than glucose tolerance.
odd font change in line 110-11 needs to be fixed
mention that in vitro studies have the caveat that they lack neural innervation and vascular supply, which are clearly important for BAT (line 115-19)
Titles of subsections are a bit misleading and should be made more consistent - ie: "cold increases fatty acid uptake and metabolism in BAT" vs. "exercise reduces lipid content in BAT" -- shouldn't the same things be explored for BAT with both interventions? Isn't lipid content driven by uptake, oxidation, etc.?
Has lipid uptake itself been measured in BAT for cold vs exercise? (ie: using a tracer? this has certainly been done in humans)
exercise section on lipid content in BAT has numerous inaccuracies - please revisit the papers of Cinti, Canon and Nedergaard, Spiegelman, etc. Cold does increase fatty acid release from WAT as well. Also cold DOES decrease lipid droplet size in BAT - this is repeatedly shown histologically.
Mitochondrial activity sections are glaringly missing seahorse data - there are many studies on this in the literature.
Batokine section has nice detail on certain factors, which is fine, but since so many are left out of the discussion there should at least be a small section that describes the other batokines that have been reported but are not discussed in detail - especially if they affect the lipid and glucose pathways which are the focus of this review
In the conclusion, it should be made more clear (perhaps in title as well!) that activating BAT likely has more promising anti-diabetes effects than anti-obesity, especially since appetite will be stimulated by BAT activation (as with exercise). Inducible/beige cells should be mentioned, especially since most human BAT is this type.
As for cold/exercise combo as a therapy, please read/cite recent commentary by Cypess and Kahn
Author Response
We thank the reviewer for their thorough review and comments on our manuscript. We have addressed all the comments in our response below and in the text of the manuscript.

Reviewer 2 Report
Carmen Peres Valgas da Silva et al. discuss the current literature comparing the effects of cold exposure and exercise on BAT activity and metabolism. The Authors concluded that cold exposure and exercise display opposite responses in BAT metabolism (glucose and lipid uptake and storage) but similar effects on batokines secretion. These conclusions are not clearly and completely supported.
However, many similar works have already been published, also from the same Kristin Stanford; in my opinion, the main strength is the idea to make a systematic, comparative analysis on the effects of cold and exercise on the different BAT functions.
However, the review will benefit if the Authors considered that:
1) the two physiological stimuli (cold and exercise) have a different metabolic/hormonal status.
2) The inconsistencies between many studies on exercise may be due to the type of exercise.
Many contradictory findings might be explained by the vast differences in the mode (treadmill, swimming, wheel running) and the experimental protocols (intensity and duration of the exercise) used. Consequently, different BAT responses and adaptations are also possible.
3) As well as during cold stimulation, sympathetic tone is increased in BAT of exercise rats. I would like to suggest the authors to consider the study of De Matteis et al, NMCD 2013. An increased parenchymal neuro-vascular network was shown in BAT of both acute- and chronic exercised rats. This could be very important to explain some data about the batokines (see line 312-315).
line 67-70 in accordance with above said, the sentence must be changed as well as the bibliographic reference.
A few points I would like to be addressed:
line 83-87: " BAT is heavily reliant on glucose as a fuel source.." and "Cold exposure and exercise have opposite effects on glucose uptake and metabolism in BAT"
line 98-99:."Studies in humans have shown that cold exposure... increase glucose uptake"
The conclusions presented in the current study may be valid for BAT, and in-line with the already reported physiological findings in some papers but not all. All these sentences as well as all the section need to be reconsidered in the light of a more in-depth examination of the literature (for a overview, please see A. C. Carpentier Frontiers in Endocrinology, 2018).
In the light of the recent characterization of skeletal muscle and BAT energy metabolism, the role of skeletal muscles in cold-induced thermogenesis and glucose metabolism in humans need to be considered (see Marlatt et al, 2018). A larger role for skeletal muscle rather than BAT in the response to cold exposure need to be considered (please, see Blondin et al J Physiol 2015).
line 135-139: the sentence "cold exposure increases BAT glucose uptake in order to provide fuel for thermogenesis" is not completely correct. I would like to suggest that BAT glucose uptake can be disconnected from thermogenesis in animals and humans (see Weir et al, Cell Metabolism 2018; Cypess et al, J Nucl Med 2013; Hankir et al, J Nucl Med 2017; Olsen et al Mol Metab 2017).
line112-114: to support the data, I would like to suggest that no change was found for GLUT 1 and GLUT4 mRNA in BAT of 1 week-exercised rats (De Matteis et al, NMCD 2013)
Line 164: What do you mean by this subtitle "3.2 Exercise reduces lipid content in BAT" without lipolysis? Your conclusion is that exercise reduces lipid uptake, accumulation and lipolysis in BAT(line 174). How do the Authors explain the data about lipid droplets remodelling and PLIN protein content (expecially PLIN5) in chronic exercised rats (SV Ramos et al, Adipocyte 2016)? What about the initial lipid droplets content in brown adipocytes?
line 272-274: "Exercise increases specific acyl chains 36:2 in phosphatidylcholines (PC), 40:5 and 40:6 acyl chains in phosphatidylethanolamine (PE), 16:0 and 16:1 acyl chains in phosphatidylserine (PS) species [90]." Can the different metabolic status (between cold and exercise) explain the specie-specific modification to the BAT lipidome?
Line 279-281: I would expect it to include a statement on these lipidomic species and I would expect it to be more focused and rounded. You have a specific competence on this interesting topic.
I agree that more work is needed to fully establish the role of exercise on the mitochondrial/thermogenic activity in BAT. However, I suggest that:
1)A first comparative analysis of BAT mitochondria of rats submitted to exercise and cold exposure (4°C) is present in De Matteis et al, NMCD 2013.
2) Recent data about the improved mitochondrial functions and energetic profile (de Las Heras et al., Frontiers Physiology 2018) need to be considered.
3)The authors conclude that" voluntary wheel running in mice decreased functional mitochondrial activity" I would like to suggest that the data showed in Lehnid et al, eScience 2019 are not conclusive.
line 212-213 "...exercise decreases the mitochondrial activity of BAT.." need to be revised.
What about UCP3?
Literature about UCPs and other mitochondrial markers are not homogeneous. Many inconsistencies between the different studies may be due to the mode of exercise (voluntary, treadmill and swimming): it's important to keep in mind this consideration!! Literature need a major analysis; (see N de Las Heras et al, Front Physiol 2018; Flouris AD et al Horm Mol Biol Clin Investig. 2017, for a systematic review.)
line 268-270 this conclusion is not clearly supported: I would suggest more prudence. It is not demonstrated that exercise inactives BAT thermogenesis.
Author Response

(The authors gave the same response as above.)

Reviewer 3 Report
This is a well-written review on the therapeutic potential of cold and exercise-induced activation of brown adipose tissue to combat obesity and diabetes. The senior author has excellent training and great expertise as well as a strong publication record in this field. This review is informative and relevant to the field. I believe this review can be even more enlightening if a few changes are made to the text. Please, find bellow my specific comments.
-Considering the debate around the characteristics of BAT in humans and how it may behave more like beige adipocytes, in addition to the fact that exercise can induce browning of adipose tissue, it may be of value to add a section on the effects of exercise on browning. Also, given that exercise changes in systemic metabolism are mediated by different organs and systems, it may be challenging to dissect the specific BAT-derived beneficial effects of exercise in humans. Therefore, experiments to test BAT-specific effects in mice should be discussed.
-On the section about the effects of exercise on mitochondrial content and activity, there seems to be a lack of discussion regarding the discrepant results observed in different independent publications. While in a couple studies exercise induced mitochondrial function, other studies show no change or a reduction in mitochondrial function. Please, discuss these contradictory results. Does it have anything do to with type, intensity or duration of exercise? Is this a reflection of the use of different approaches to measure/estimate mitochondrial function?
-Regarding the section on Batokines (FGF21): Please, add a reference demonstrating that FGF21 is dispensable for long-term cold adaptations (Keipert, S. et al; 2017) and discuss why this may be the case? Does endogenous BAT-derived FGF21 have different effects than recombinant FGF21 on BAT activity? Please, also refer to the manuscript describing that FGF21 effects on energy expenditure may be independent of its action on adipose tissue (BonDurant, LD et al; 2017).
-In regards to exercise-induced increases in FGF21, please, additional information is required regarding reference 111. Although, the source of FGF21 was not determined, their data show that FGF21 mRNA levels are elevated in the liver, but not white adipose tissue. I understand that FGF21 expression was not investigated in BAT, but their data suggest liver may be the source for circulating FGF21 in that particular study.
-What about additional batokines, such as IL6, NRG4, Angptl8 and others?
-Finally, on line 373 the authors claim that the effects of exercise and cold on batokines are similar. This claim is not well-substantiated in the current form of the manuscript. Additional literature should be added to support this claim or authors should tone the statement down a bit. In addition, limitations to using exercise, cold exposure or a combination of both as therapeutic tools to treat obesity should be discussed in future perspectives.
Minor: Please, check font type and size and it changes in a few areas of the manuscript
Author Response

(The authors gave the same response as above.)

Round 2
Reviewer 1 Report
All concerns have been adequately addressed and the revised resubmission is an outstanding review article that will likely be highly cited.
Reviewer 2 Report
The revised manuscript is now improved and should be published.